# The Immediate Effect of Informational Manual Therapy for Improving Quiet Standing and Bodily Pain in University Population

**DOI:** 10.3390/ijerph18094940

**Published:** 2021-05-06

**Authors:** Rosa Cabanas-Valdés, Mª Dolores Toro-Coll, Sara Cruz-Sicilia, Laura García-Rueda, Pere Ramón Rodríguez-Rubio, Jordi Calvo-Sanz

**Affiliations:** 1Physiotherapy Department, Faculty of Medicine and Health Sciences, Universitat Internacional de Catalunya, 08195 Sant Cugat del Vallés, Spain; lolatorocoll@gmail.com (M.D.T.-C.); lauragarcia.fisio@gmail.com (L.G.-R.); prodriguez@uic.es (P.R.R.-R.); 2PROMOVE Mairena de Aljarafe, 41927 Sevilla, Spain; fisio@saracruzsicilia.com; 3Physiotherapy Department, School of Health Sciences, Tecno Campus, Mataró-Pompeu Fabra University (TCM-UPF), 08302 Barcelona, Spain; jcalvo@tecnocampus.cat; 4Hospital Asepeyo Sant Cugat del Vallès, 08174 Barcelona, Spain

**Keywords:** standing balance, manual therapy, therapeutic touch, quiet stand, pain and health status

## Abstract

Background: The Informational Manual Therapy (IMT) is a therapeutic touch. This study aims to assess the effect of IMT on quiet standing, pain and health status in university population. Methods: An experiment was conducted on subjects utilizing a comparative paired analysis both before and after the intervention. One IMT session was performed on 57 healthy individuals aged from 18 to 65 years. The primary outcome was quiet standing assessed by the Satel 40 Hz stabilometric force platform. Secondary outcomes were bodily pain assessed by the 36-Item Short Form Survey (SF-36) and health status by EQ-5D-3L. The primary outcome was evaluated before and immediately after treatment. Results: The individuals were divided into 3 age groups, 18–35 (52.6%), 35–50 (29.8%) and 51–65 (17.6%). Statistically significant differences were immediately observed after the session ended when comparing the pre-post quiet stance scores in a number of length parameters: L, Lx, Ly and stabilometry amplitude on *Y*-axis with eyes open and closed. Significant differences were also found when testing bodily pain (SF-36) and anxiety (5Q-5D-3L). Conclusion: One session of IMT produced positive effects when testing quiet standing with eyes open and eyes closed, as well as a significant reduction in pain and anxiety for those tested. Further research is suggested.

## 1. Introduction

The Informational Manual Therapy (IMT), also known to as Poyet-Pialoux method, is a therapeutic touch [1] using a whole-body approach with light touch. IMT harmonizes the human body through the natural connection of a cranial-pelvic fascial system. This system consists of the fluid and membranes enclosing and protecting the brain and spinal cord, as well as the attached bones [2]. These membranes extend from the bones of the skull and face and down the spine to the sacrum or tailbone area [3]. It plays a vital role in maintaining the environment in which the central nervous system functions [4]. Since the brain and spinal cord are contained within the cranial-pelvic system, it can have a powerful influence over a wide variety of bodily functions [5]. The manual palpation of this system supposedly affects the sensory, motor, cognitive, and emotional processes in the nervous system [6].

Therapeutic touch involves a five step process of *centering* (generation of a quiet, focused state and the intention to heal), *assessment* of the energy field using the hands to sense energetic cues, modulation of energy to balance and restore the energy field, and an *evaluation* phase to determine the restoration of balance to the energy field of the recipient, which is purported to produce a relaxation effect [7,8]. A relaxation response refers to positive physiological changes induced by an autonomic response that diminishes the hypothalamic-pituitary-adrenal axis response of the autonomic nervous system by reducing norepinephrine release [8]. Neuroscientific evidence has demonstrated that hands on therapies have positive emotional effects on humans. The role of touch in humans and its fundamental meaning for social animals have been well established in neuroscience [9,10]. An identifying feature of the IMT is the use of skull somatotopies [11] like that are performed in reflexology [12] or auriculotherapy [13]. These somatotopies are reflex points located on the scalp and face, where every region of the body is represented. The IMT diagnosis is performed on the skull and the treatment is performed mainly on the sacrum.

According to the literature, IMT focuses on optimizing the correct motion of the secondary respiratory mechanism (SRM), also known as cranial rhythmic impulse [14]. SRM is a slow involuntary rhythmic pulsation palpable on the external surface of the head of a living human subject and where the rhythms of the skull depend on cardiac and respiratory activity and cerebrospinal fluid [15]. It is transmitted through the fascia and the muscle chains to every system and body part [16,17,18]. This movement is due to the fluctuation of the cerebrospinal fluid in its tour along the bone marrow, from the skull to the sacrum [19].

The movement of the brain and spinal cord, cerebrospinal fluid, meninges, and bones are all synchronous with each other, forming one large integrated unit of function [20] from the movement of the brain and the spinal cord; a movement which would create outward tension waves slower than that of the real rhythm of breathing and heartbeat, thanks to the viscoelastic property of the meninges that dampen the speed of tension transmission [21]. The SRM is based on the movements of the brain mass induced by the myocardium and the diaphragm muscle [22], which, through the mechanical properties of the brain, neurofluids [23], meninges, and the bone suture complex [24], palpation detects cardiovascular and respiratory health. The contact between patient and therapist, from a field dynamics point of view [25,26], creates a bi-directional meeting of magnetic information [27].

At the SRM level, inertia within the system may be perceived as tissue resistances within and between the separate structures. It spreads to the entire body structure and tissue through the fascial or connective system. The fascial tissues may also be seen as an essential extension of the dura mater (meningeal) [28]. Fascial alterations further inhibit weaker muscles, with stronger muscles becoming hypertonic and indicative of implicit, unresolved biopsychosocial issues [29,30]. In a healthy body, the fascial system is free to move and transmit tension along the path. However, in case of rigidity of the fascial structures, elements will not be in balance, resulting in pain and possibly eventual injury [31,32]. Any functional or structural alteration of the SRM will be reflected in any part of the body since it behaves as a unit [33]. SRM is measurable and is palpable by a hand therapist throughout the body [34,35].

The body is made up of many parts which relate to each other in various ways. Altered muscle tension disturbs the normal mechanical behavior or “function” of the area and is part of the body’s response to injury which is known as “somatic dysfunction”. The stability of the body therefore seriously risks being altered. Bipedal vertical posture is inherently unstable in the gravity field [36]. Normal standing is a complex activity both mechanically and neurologically in that the apparently simple act of standing upright involves complex, dynamically regulated sensorimotor integration mechanisms [37]. One of the goals of sensorimotor integration for postural control is to ensure that an adequate amount of corrective torque is generated to resist the destabilizing influence of gravity and other external perturbations. Balance control involves the integration of various inputs and systems, enabling the individual to remain upright by controlling the relationship between the center of mass and the base of support [38]. For the proprioceptive system, a number of different sensory receptors (muscle stretch receptors, joint receptors such as golgitendon organs, Ruffini terminals, Pacinian corpuscles, foot pressure and free nerve endings) contribute to an overall sense of body motion relative to the feet. For modelling purposes, we assume that the nervous system responds to the complex array of sensory cues that occurs with body sway and is able to extract a signal that accurately encodes ankle joint motion over a wide dynamic range [39]. According to Philippi et al. [40], manipulative therapy has an influence on postural asymmetry. In a study of Tai Chi training [41], the individuals presented much better postural stability compared to the general populations.

The hypothesis of this study is that IMT has an influence on postural control, bodily pain and health condition. As a consequence, the aim of the study is to analyze the effectiveness of IMT as an intervention for the lessening of pain, and improvement of quiet standing balance and health status in university population with somatic dysfunctions.

## 2. Materials and Methods

### 2.1. Study Design

The study utilizes a one-group pretest-posttest design. The study protocol was approved by the Human Research Ethics Committee of the International University of Catalonia (UIC) with the number: FIS-2019-02 pursuant to the Declaration of Helsinki. The personal data of individuals remained confidential, and the data were shared anonymously upon request submitted by the principal researcher. The TREND statement was followed [42]. The study was registered in ClinicalTrials.gov identifier with the number: NCT04404829.

### 2.2. Subjects

Fifty-seven participants were recruited to participate into this study; the sample was one of convenience. The recruitment document explained that participation was voluntary, without incentives offered to participants and dependent on the inclusion and exclusion criteria. The inclusion criteria were as follows: individuals ranging in age from 18 to 65 years old with somatic dysfunctions, devoid of a positive diagnosis for any form of disease which influences balance, who were not participating in any other trial, and who were free of injury in the 3 months prior to the study, with no fractures in the previous 6 months, and not having suffered any falls in the preceding month. Exclusion criteria were any individual not meeting all of the above mentioned inclusion criteria. No subject had been under any pharmacological treatment during the previous 4 weeks. Every individual provided informed consent before enrolment.

### 2.3. Intervention

Every participant received one single session of IMT. All the treatment sessions took place at the UIC laboratory. Room temperature was always maintained at 29.8–34.5 °C and relative humidity at 39–42% (Oregon Scientific Model pe 299N; Ltd., Maidenhead, UK) and without any noise. The IMT session was performed over two days by four physiotherapists and a physician, all having more than five years of experience in IMT treatment.

The first level of IMT was applied in this study. The objective is to harmonize the cranial-pelvic system, as this is essential for maintaining a correct posture. The first step consists of balancing the “fuses,” vibration areas that act as a basic protection system for the body. These local spots cease functioning at times, thus resulting in an interruption of the correct SRM: First step, 1st fuse: Inn-trann, (Figure 1) the therapist places her/his index fingertip on the forehead, between the eyebrows, at the base of the glabella. If not balanced, the therapist does the adjustment between spines 4th and 5th sacral vertebrae. A circular movement on the scalp in several very soft circles that shifts from the frontal to the occipital area is performed 6 times (Figure 2). 2nd fuse: 1st dorsal, 4th cervical, 4th dorsal and 3rd lumbar vertebra on the spinous process. To check the one hand is placed on the lumbar vertebrae and the other on dorsal vertebrae (Figure 3). The therapist should notice a subtle synchronic cranio-caudal movement. 3rd fuse: C0 (space between occipital and 1st cervical spinous process), 2nd sacral vertebra and the space between the spinous process of the 2nd and 3rd dorsal vertebrae. One hand is placed on the sacral vertebrae and the other on cervical vertebrae to check the SRM movement. If any point is blocked, it is treated and harmonized with a soft touch towards the occipital.

Second step, to release the axes of the sacrum and sacroiliac joints to free the pelvis and promote its correct functioning.

### 2.4. Outcome Measures

The primary outcome was static standing balance. It was assessed using the Satel 40 Hz stabilometric force platform model PF2002, software v 33.5 8C (France) [43]. All the participants underwent a force platform measurement before the intervention (T0), immediately after intervention (T1), and 7–10 days following the end of intervention (T2). Device calibration was performed prior to each measurement. The participants were instructed to quietly stand upright on the force platform, and to look at a red colored small object (1.5 × 1.5 cm) placed on a white screen, at eye level, at a distance of 90 cm in front of the force platform. They stood barefoot with their arms along their trunk, and their feet were parallel (heels separated by 2 cm and of external hip rotation of 30°) as recommended by the *French Posturology Association-Rule 85*. They were not allowed to speak or tighten the temporomandibular joint as this influences the assessment. The data was first collected with the subject in the eyes open (EO) condition for 51seconds, and then with eyes closed (EC) for the same time duration. Two trials were performed with EO and EC, alternately and consecutively for everyone. The stabilometric test was conducted under the same environmental conditions for all subjects.

Secondary outcomes were bodily pain using the Medical Outcomes Study Short Form-36 (SF-36) and health-related quality of life using the EQ-5D-3L and health status by EQ-5D. These questionnaires were administered at T0 and at 3 weeks. The SF-36 answer items used in this study referred to the past 4 weeks. Only the pain sections were used: *How much bodily pain do you have?* and *How much did pain interfere with your normal work?* (including both work outside the home and housework). The EQ-5D-3L consists of 2 pages, and the descriptive system comprises five dimensions: mobility, self-care, usual activities, pain/discomfort and anxiety/depression. Each dimension has 3 levels: no problems, some problems, and extreme problems and a visual analogue scale for health ranging from 0 (worst possible) to 100 (best possible).

These measurements were performed by a physiotherapist who was not involved in the treatment.

### 2.5. Data Analysis

Absolute frequencies and percentages were used to describe categorical variables. The mean and standard deviation was used to describe the numerical variables of a normal distribution. The median and the first-third quartile were employed if the variable did not follow a normal distribution. The Student-t test contrast was adopted if the data followed a normal distribution and the Mann-Whitney U test when the distribution was not normal. The Student-t test or Wilcoxon contrast is used to analyze the change in a continuous variable over time, and for paired data, depending on the normality of the variable to be analyzed. Finally, the X2 contrast is used to analyze differences in categorical variables if the data are independent, and McNemar’s test if they are paired. The Shapiro Wilk test was applied in order to check normality. The analyses were performed with the R 3.6.1 software package, with 95% confidence interval and a significance level of 5% (*p*-values < 0.05).

## 3. Results

The Table 1 summarizes of individual’s characteristics.

In the force platform assessment, all the individuals performed T0 and T1, and 41 of them performed the T2. All the individuals answered the questionnaires and personal information at T1, and 49 did so at T2 (3-week) (Figure 4). Whether they habitually engaged in any physical activity was recorded.

There were no statistically significant differences between the five therapists who performed the intervention. Statistically significant differences were observed in the parameters referring to values of length, such as L, Lx and Ly, with EO and EC, as well as in the parameter stabilometry amplitude in Y with EO at T1 and between T1 and T2 (Table 2) and (Figure 5, Figure 6 and Figure 7). There were statistically significant differences between EO and EC, meaning that individuals fell within the normal range. The age range that improved most was 18–35 years. Women presented a greater balance with EO, in L and Ly parameters than men.

The SF-36 items for the statistical analysis of bodily pain, were grouped into: “none, moderate and severe”. The pain interference item was grouped as “not at all, moderately and extremely”. Statistically significant differences were found for the two items in the population that carried out physical activity and also in the EQ-5D-3L (Table 3). Statistically significant differences were found for pain in subjects that did physical activity (Table 4). Statistically significant differences were reported for the anxiety item and the visual analogue scale (EQ-VAS) (Table 5). No adverse events were observed.

## 4. Discussion

This study showed that only one session of IMT brought about immediate positive effects on quiet standing with EO and EC in individuals with somatic dysfunctions. IMT reduces anxiety/depression and bodily pain at three weeks. To date, there is no scientific evidence about the effect of IMT, and it is not possible to compare our results with any other study. The physiological mechanisms of action of IMT are not entirely known. One of the hypotheses currently under consideration is that IMT activates the body’s self-regulating elements which would help its balance in all aspects.

After a session of IMT the neuromuscular system and postural control system were more efficient because the postural sway had decreased both with EO and EC. In our opinion the fascial system had more freedom to move and transmit the tension to the tendons more rapidly [30]. It may hence be assumed that the insertions allow for the selective tensioning of the fascia, which in turn, supports and facilitates muscle activity via muscle and fascia mechanical interactions. Muscles forces, acting directly or indirectly on the fascial system via muscle-fiber attachments, may manipulate not only muscle function, but also proprioception [21]. Structural continuity is not limited to the linkages between the fascia and the muscle. The connective tissue surrounding the adjacent compartmental muscles of the lower limb has been shown to be tightly fused [17]. Likewise, there is also evidence for myofascial continuity in an in-series arrangement: a fascial band connects the proximal end of the gastrocnemius to the distal semitendinosus [44]. In view of the intimate morphological relationship between muscles and associated connective tissues, the effects of local stiffness changes and altered local muscular forces might affect both the tissue of origin and the surrounding areas.

This hypothesis relays on the concept of biotensegrity [45,46]. Tensegrity is an architectural system where structures stabilize one another by balancing the counteracting forces of compression and tension, thus shaping and strengthening both natural and artificial forms [47]. In the biological field, the principle of continuous tension or discontinuous compression may also be used to demonstrate the structural integration of the body [48]. The muscular and bone system could be seen as a system of biological tensegrity. Body fluids or liquid fascia influence the shape and function of the cell and organs, causing solid fascia to adapt through mechanical transduction. The sacrum receives the load form the spine and this is transferred to the hips through compression forces transmitted along the ilium into the acetabulum [49,50,51]. According to the conventional arch model of the pelvis, stability of the sacroiliac joints comes from compressive forces of transversely oriented fibers of ligaments and muscles (force closure) [52].

Biotensegrity can be demonstrated at all scales in the human body [53]. From molecules to tissues and organs, each level can be viewed as a biotensegrity structure, intimately connected with the level above and below in a hierarchical organization [54]. This may explain how forces applied through the skin during IMT session may have effects at the cellular level. Since the fascial chains of the body attach onto the cranium [3] these forces may have an effect on the fascial system, where the external fascia chains are related to posture, and internal fascia chains to supporting functions.

It would be well-advised to use and deepen on the term fascintegrity on constructing a new model to gain an understanding of the adapting behavior of cells and tissues, as well as musculoskeletal movement. The integrity is given by the fractal and entropic organization that from the cell reaches the epidermis, whose structures, solid and liquid, are fasciae [55]. Muscles play a valuable role in managing the mechanical tension produced and felt by rapidly changing the morphology of their cytoskeleton; this mechanism is facilitated by the intervention of fibroblasts [32]. If the mechanical stimulation felt by the myofascial system (connective and contractile tissue) is present for a short period, the morphological change will be transient. If the mechanical forces persist in reshaping the myofascial system, this will result in a chronic change in form and function [21].

IMT seems to be induced by delicate stimulation of the mechanoreceptors in the fasciae, which can cause changes in the autonomous nervous system, leading to inhibition of sympathetic activity and increased parasympathetic activity. Therefore, due to the existence of a two-way interaction between the activity of the autonomous nervous system and the fascial tonicity [31], IMT may liberate to the fascial constraints in all body areas by regulating both the autonomous system and the fascial tonicity.

In addition, the higher centers of the central nervous system are involved in different aspects of postural control and balance [38]. Indeed, postural response has been shown to be modified by various cognitive-motor processes ‘represented’ in the cerebral cortex, including those involved in changes in cognitive load and attention when performing dual postural tasks. At the end of the IMT treatment, the participants were more relaxed, and their neuromuscular activity had decreased with a decline in sympathetic activity, which may had contributed to their improved quiet stance. According to Pan et al., after Tai Chi training [41] the joints of individuals were slightly flexed and their muscles relaxed. This means that excessive muscle co-activation is not good from the point of view of postural stability in the field of gravity. The IMT promotes a relaxed body and mind. This is an important aspect for maintaining an equilibrium between excitation and inhibition as far as responding and adaptation to environmental demands are concerned [56]. This therapy acts on the neurovegetative and endocrine system which explains the feeling of well-being, calmness or relaxation experienced by every individual after a IMT session [8,57]

If IMT improves the precision of the fine postural system, it may prove to be an important aspect in individuals with postural impairments. Further research is recommended to support these findings. The importance of this field of research becomes clear when considering that a deterioration in balance results in a seriously impaired quality of life [58].

Our participants, consisting of a university population, had stated positive effects regarding bodily pain and anxiety. They were healthy adults with somatic dysfunctions and were not diagnosed with any disease. Of the participants, 85% of them presented bodily pain and 38% anxiety. This was in line with the results of a meta-analysis [59]. They showed that a 33.8% of medical students suffered anxiety. Backåberg et al. [60] revealed that university education has an impact in musculoskeletal symptoms. Our results suggested that this could be an important factor in relation to patients with various diseases and individuals suffering from stress. With regards to stress, the cerebral cortex sends signals to the autonomous nervous system, turning the sympathetic to becoming active [61]. This results in the increase of cardiac frequency and changes in the variability of the frequency, reducing the flow of peripheral blood and the flow of renal blood. This increases the blood pressure and vascular resistance.

There are several similitudes between IMT and osteopathy. Both are a therapeutic touch approaches aimed at enhancing self-regulation of the body, focusing on somatic dysfunction correction. A study by functional magnetic neuroimaging (fMRI) showed that osteopathy has a positive effect on brain activity, particularly on the insula [62]. The insula has an influence in pain perception. The fundamental significance of the insula is its ability to serve as a subjective experiential and feeling center that integrates emotional, sensory, cognitive, and motor functions. Cerritelli et al. [62], using fMRI, demonstrated that osteopathy induces lasting blood oxygen level-dependent effects on crucial areas of interoceptive networks in patients with low back pain. Another fMRI study showed perfusion changes in a cortical area involved in the dynamic balance between the sympathetic and parasympathetic systems, which would suggest a corrective effect of osteopathy on the disruptive influence that biomechanical strains have on those systems [63]. The innervation that affects the fascial system is autonomous: sympathetic and parasympathetic. Tramontano et al. [64] provides the first preliminary evidence of brain network connectivity changes resulting from osteopathic manipulative treatment.

The fascial system is constituted the same way as that nerve structure. All layers are innervated and have a thin but potentially important plexus of nociceptors [2]. The muscular system is part of the fascial continuum, and in the presence of systemic diseases and disorders of the visceral, genetic, vascular, metabolic and alimentary type, it undergoes a non-physiological alteration of its function [65]. Epigenetic processes lead to adaptation in response to a lack of mechanotransductive, causing a further decline in its properties [66].

Potential IMT effects on pain and anxiety could be caused by a placebo effect [8,67]. For this reason, we used the posturo-stabilometric examination. In our opinion, the stabilometry allows for an objective study of body sway during quiet standing, in the absence of any voluntary movements or external perturbations. In our opinion, this is not the effect of learning and placebo.

### Limitations

The main limitation of this study is the convenience sample, the lack of control group and the small sample size.

## 5. Conclusions

One session of IMT produced immediate positive effects on quiet standing with eyes open and closed conditions. IMT reduces bodily pain and levels of anxiety in healthy individuals on a short term but may have a placebo effect. The study of manual therapies from a neuroscientific point of view could well be expected to lead to fresh insights in a field of research that remains unexplored, despite the widespread use of different manual approaches and touch-based interventions [62]. Additional studies are now required in order to gain a comprehensive understanding of the key tenets of IMT. Future randomized studies of IMT therapy are currently needed, involving people affected by a disease and suffering from pain, anxiety and disturbances of balance, such as vestibular syndrome or fibromyalgia.

## Figures and Tables

**Figure 1 ijerph-18-04940-f001:**
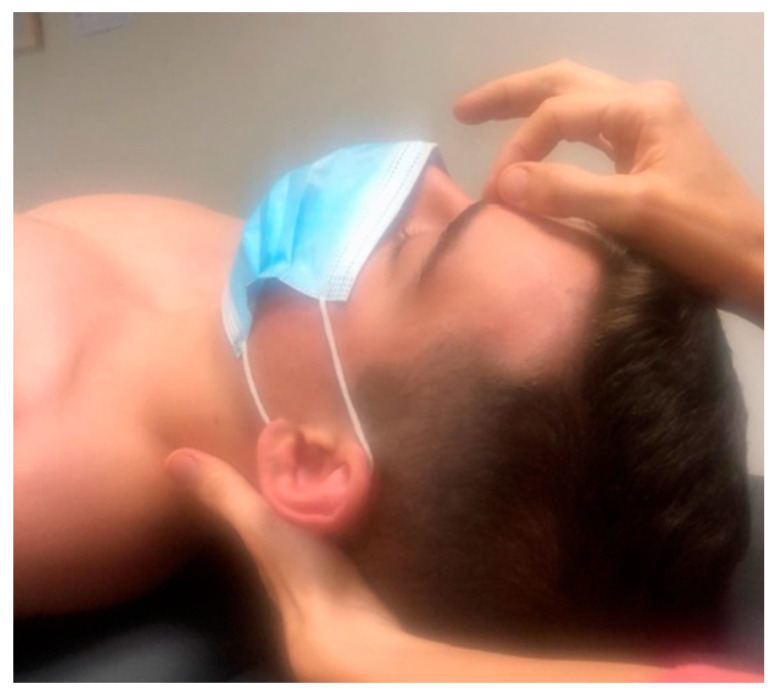
Checking 1st fuse: Inn-trann.

**Figure 2 ijerph-18-04940-f002:**
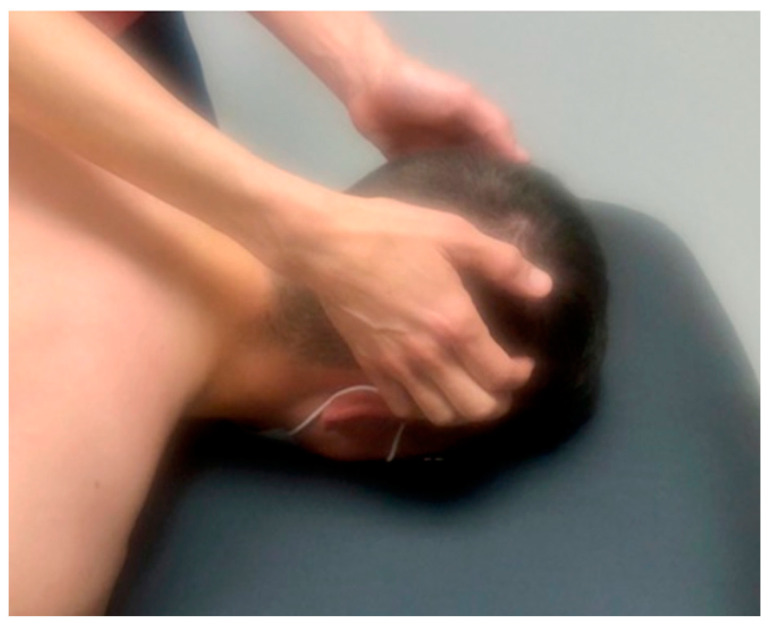
Circular movement on the scalp.

**Figure 3 ijerph-18-04940-f003:**
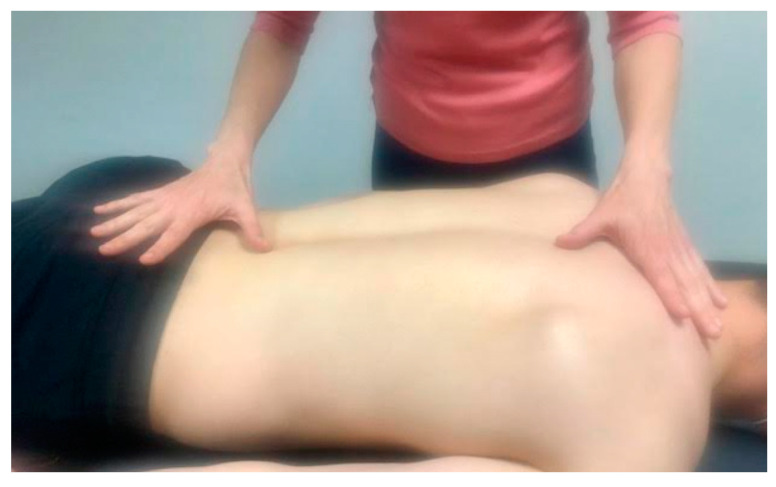
Checking cranio-caudal movement.

**Figure 4 ijerph-18-04940-f004:**
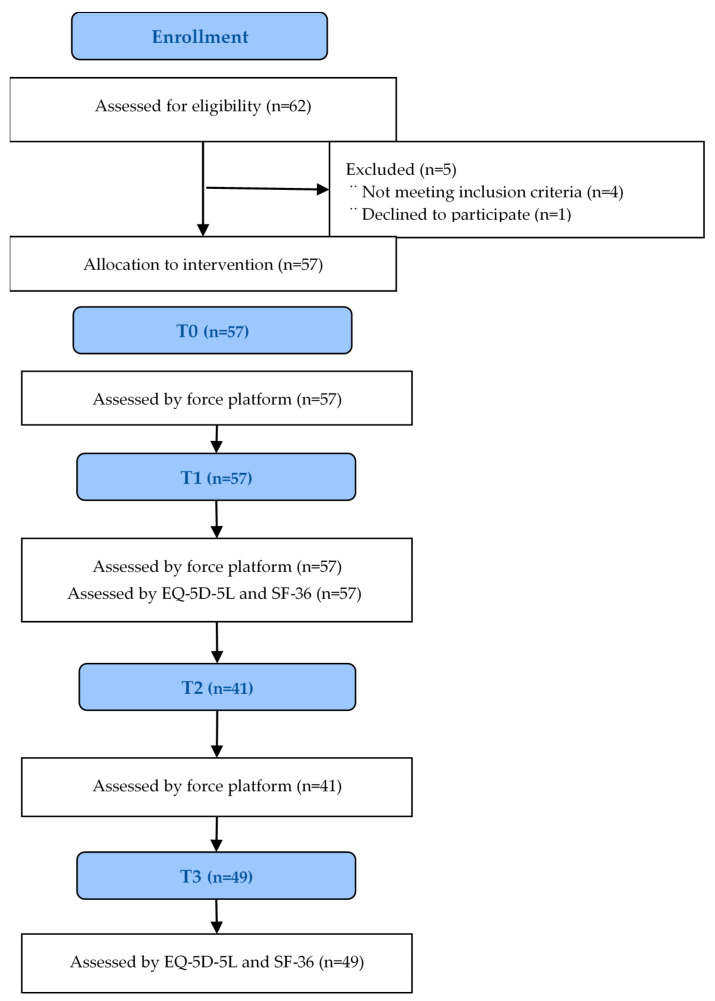
Flow Chart of the study.

**Figure 5 ijerph-18-04940-f005:**
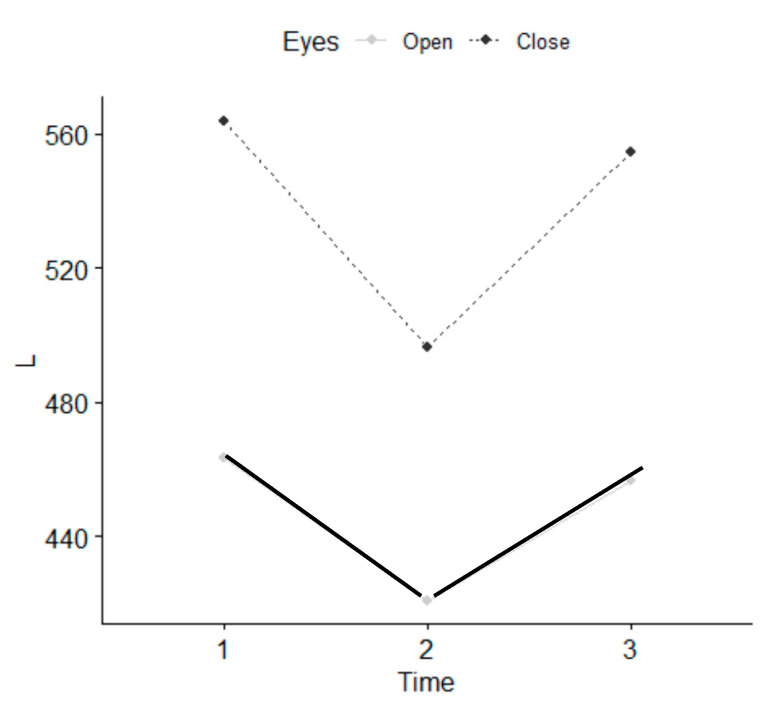
Length (L) parameter, 1 (T0 before), 2 (T1 immediately), 3 (T2 7–10 days).

**Figure 6 ijerph-18-04940-f006:**
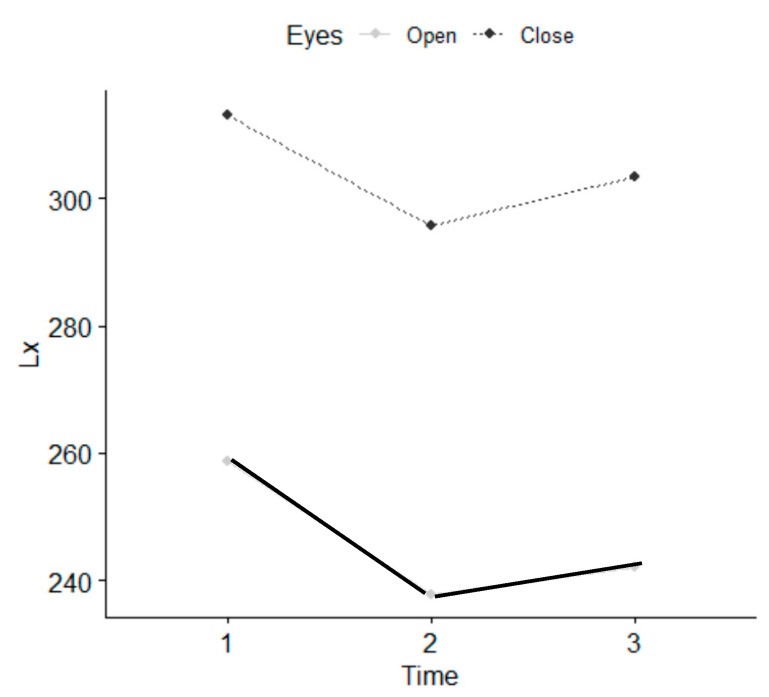
Length in (Lx) parameter 1 (T0 before), 2 (T1 immediately), 3 (T2 7–10 days).

**Figure 7 ijerph-18-04940-f007:**
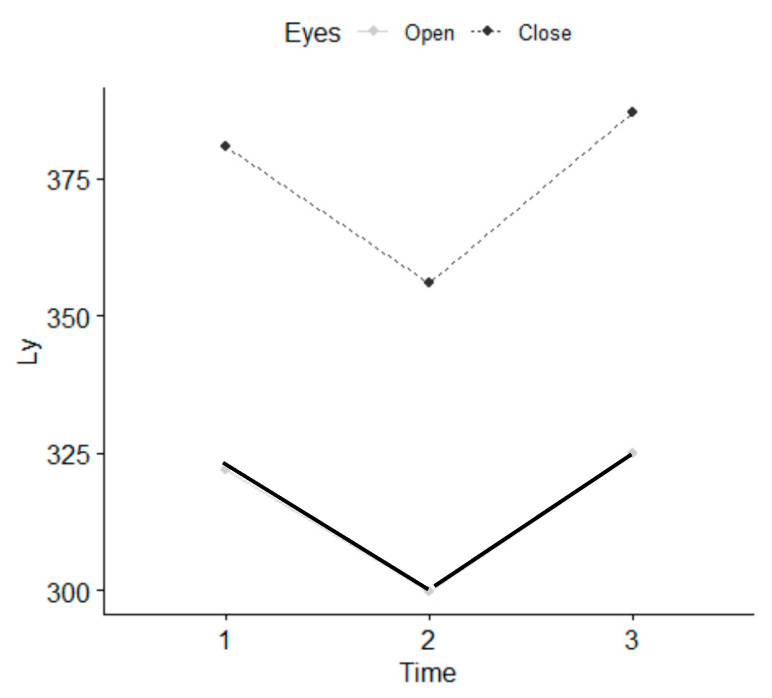
Length in (Ly) parameter, 1 (T0 before), 2 (T1 immediately), 3 (T2 7–10 days).

**Table 1 ijerph-18-04940-t001:** Summary of individual’s characteristics.

	Mean (SD)	Median (1–3Q)	
Age	35.1 (13.1)	33 (24–45)	
The mean length of their right foot	39.9 (2.9)	39 (38–42)	
Height	1.7 (0.01)	1.7 (1.6–1,8)	
Weight	66 (14.9)	63 (55–74)	
Mean body mass index	28.8 (7.4)	26.8 (33.9–43,2)	
Age group	Total	Women	Men
18–35 years	30 (52.6%)	18 (48.6%)	12 (60%)
35–50 years	17 (29.8%)	11 (29.7%)	6 (30%)
50–65 years	10 (17.6%)	8 (21.6%)	2 (10%)

SD: standard deviation.

**Table 2 ijerph-18-04940-t002:** Comparison between pre-post intervention with eyes open and closed.

	Pre (T0) n: 57				Post (T1) n: 57				*p*-Value
	Mean (SD)	SEM	Median (Q1–Q3)	CV(%) *	Mean (SD)	SEM	Median (Q1–Q3)	CV(%) *	
**Eyes Open**									
Surface	245 (98.7)	13.1	242 (172.7–310.4)	40.3	256.5 (148.4)	148.4	217.3 (154.6–304.5)	57.9	0.975
Xm	3.1 (6.4)	0.8	3 (−0.2–7.4)	206.5	3.4 (6.1)	6.1	3.5 (0.3–7.2)	179.4	0.606
Ym	−46.9 (12)	1.6	−45.2 (−53.4–38)		−46.4 (9)	9	−46.8 (−52.4–38.4)		0.997
Length	500 (142)	18.8	463.4 (400.1–592.8)	28.4	470.2 (138.7)	138.7	421 (381–550.5)	29.5	**0.002**
Lx	275.5 (80.8)	10.7	258.7 (222–327.9)	29.3	256.6 (75.8)	75.8	237.8 (212.2–305.2)	29.5	**0.004**
Ly	357.5 (110.3)	14.6	322 (278.2–425.7)	30.9	338.8 (107.5)	107.5	299.8 (258.2–389.3)	31.7	**0.003**
s X maximum	12.5 (6.7)	0.9	13.7 (8.4–16.9)	53.6	12.4 (6.6)	6.6	13.4 (8.9–16)	53.2	0.987
s X minimum	−6.7 (6.7)	0.9	−7.3 (−10.6–2.7)		−6.1 (6.2)	6.2	−5.3 (−9.8–2.2)		0.489
s X amplitude	19.2 (5)	0.7	18.6 (16–23.2)	26	18.4 (5)	5	18.3 (14.1–21.1)	27.2	0.089
s Y maximum	−34.3 (12.1)	1.6	−34.4 (−41–27)		−34.7 (11)	11	−33.2 (−43.7–27.6)		0.546
s Y minimum	−60 (13.3)	1.8	−59.5 (−68–50)		−59.1 (12.3)	12.3	−57 (−66.8–49.8)		0.187
s Y amplitude	25.6 (6.3)	0.8	25.7 (21.5–29.5)	24.6	24.4 (6.2)	6.2	23.1 (20.3–29)	25.4	**0.04**
**Eyes Closed**									
Surface	229.7 (107.4)	14.2	223.4 (144–280.1)	46.8	234.7 (147.5)	147.5	177.5 (131.4–310.5)	62.8	0.849
Xm	3.7 (7)	0.9	4.5 (−1.6–8.6)	189.2	3.5 (6.4)	6.4	3.6 (0.2–7.5)	182.9	0.927
Ym	−44.9 (12.6)	1.7	−44.3 (−48.9–36.8)		−46.6 (9.1)	9.1	−45 (−53–39.5)		0.074
Length	617.5 (207.1)	27.4	563.7 (462.1–772.2)	33.5	564.4 (209.1)	209.1	496.2 (418.3–714.8)	37	**0**
Lx	338.1 (114.4)	15.2	313.1 (245.1–446.4)	33.8	307.9 (112.3)	112.3	295.7 (218–393.4)	36.5	**0.002**
Ly	442.3 (164.3)	21.8	380.9 (319.8–565.2)	37.1	405.4 (161.9)	161.9	356.1 (301.9–501.5)	39.9	**0.001**
s X maximum	13.5 (7.3)	1.0	14.7 (10.9–18)	54.1	13.2 (7.3)	7.3	13.2 (9.7–17.8)	55.3	0.769
s X minumum	−6.9 (7.6)	1.0	−8.1 (−11.8–0.3)		−6.7 (7.7)	7.7	−6.7 (−11.1–2)		0.432
s X amplitude	20.5 (6)	0.8	20.3 (15.8–26.2)	29.3	19.9 (7.3)	7.3	18.2 (14.4–24.9)	36.7	0.578
s Y maximum	−32.9 (13.4)	1.8	−33.1 (−38.6–24.7)		−35 (10.2)	10.2	−32.6 (−43.8–27.8)		0.078
s Y minimum	−57.1 (12.8)	1.7	−54.4 (−63.2–49.4)		−58.7 (10.1)	10.1	−58.2 (−63.9–51.2)		0.053
s Y amplitude	24.2 (7)	0.9	22.6 (19.4–28.7)	28.9	23.7 (9.1)	9.1	21.2 (17.5–27.3)	38.4	0.436

CV: coefficient of variation, s: stabilometry, SD: standard deviation, SEM: standard error of measurement, Wilcoxon test, * Not calculated if mean is negative. Bold: statistically significant.

**Table 3 ijerph-18-04940-t003:** Comparison between first and second post intervention with eyes open and closed.

	1st Post (T1) n: 41				2nd Post (T2) n: 41				*p*-Value
	Mean (SD)	SEM	Median (Q1–Q3)	CV(%) *	Mean (SD)	SEM	Median (Q1–Q3)	CV(%) *	
**Eyes Open**										
Surface	245 (149.2)	23.3	196.5 (147.1–272.6)	60.9	225.8 (99.4)	35.3	194.3 (152.1–278.3)	44	0.361	
Xm	3.9 (6.2)	1.0	4.7 (0.9–7.8)	159	2.7 (5.8)	0.4	2.7 (−0.8–5.4)	214.8	0.184	
Ym	−45.5 (8.5)	1.3	−44.8 (−51–38.4)		−46.1 (12.8)	−7.2	−44.3 (−55.1–37)		0.411	
Length	459.3 (142)	22.2	408.8 (366.2–528.2)	30.9	501.8 (145.2)	78.4	456.7 (410–558.2)	28.9	**0.001**	
Lx	246.8 (72.7)	11.4	218.3 (204.3–303.7)	29.5	269.8 (75.3)	42.1	242 (221.9–331.5)	27.9	**0.013**	
Ly	334.8 (112.7)	17.6	299 (254–377.2)	33.7	363.7 (119.3)	56.8	324.8 (290.9–409.9)	32.8	**0.002**	
s X maximum	12.3 (6.6)	1.0	13.2 (9.6–15.9)	53.7	11.9 (5.6)	1.9	11.8 (10.1–15.5)	47.1	0.672	
s X minimum	−5.3 (6.1)	1.0	−4.8 (−9.4–1.4)		−6.1 (6.5)	−1.0	−7.3 (−10.9–1.5)		0.441	
s X amplitude	17.6 (4.8)	0.7	18 (13.6–20.4)	27.3	18.1 (4.6)	2.8	17.5 (14.4–21.2)	25.4	0.418	
s Y maximum	−33.3 (11.5)	1.8	−31 (−41.8–27)		−33.6 (13.3)	−5.2	−32.5 (−44.2–23.8)		0.808	
s Y minimum	−57.7 (11.8)	1.8	−55.8 (−65.1–47.7)		−58.5 (14.3)	−9.1	−58.3 (−67.3–48.4)		0.758	
s Y amplitude	24.4 (5.6)	0.9	24.2 (20.6–29)	23	24.8 (7.1)	3.9	22.4 (20.6–28.7)	28.6	0.837	
**Eyes Closed**										
Surface	226.3 (152.9)	23.9	166 (120.7–264.7)	67.6	224.3 (108.6)	35.0	196.8 (126.5–335.8)	48.4	0.72	
Xm	4.2 (6.6)	1.0	4.1 (1.1–8)	157.1	3.3 (6.2)	0.5	4.3 (−1.4–7.4)	187.9	0.202	
Ym	−46 (8.9)	1.4	−42.3 (−51–39.3)		−45.1 (11.8)	−7.0	−45.3 (−50–37.1)		0.778	
Length	558.3 (227)	35.5	486.9 (410.1–639.1)	40.7	607.6 (194.2)	94.9	554.7 (482.1–740.4)	32	**0.016**	
Lx	301.8 (117.8)	18.4	269.8 (202.9–379.8)	39	326.7 (102.7)	51.0	303.3 (245.5–393)	31.4	0.069	
Ly	403.8 (175.9)	27.5	353.5 (300–465.3)	43.6	441.5 (155.5)	69.0	387.1 (342.8–496)	35.2	**0.019**	
s X maximum	13.5 (7.1)	1.1	13.8 (10.1–17.8)	52.6	12.9 (6.9)	2.0	14.2 (8.6–17.1)	53.5	0.48	
s X minimum	−5.7 (8.3)	1.3	−5.2 (−11.1–0.7)		−6.8 (7)	−1.1	−6.2 (−12.7–2.5)		0.252	
s X amplitude	19.2 (7.4)	1.2	17.1 (14.3–22.7)	38.5	19.7 (6.4)	3.1	18 (14.7–23)	0.3	0.99	
s Y maximum	−34.4 (10.3)	1.6	−32.6 (−42.3–26.5)		−33.4 (12.5)	−5.2	−32.5 (−40.2–24.4)		0.581	
s Y minimum	−57.8 (10.1)	1.6	−55.8 (−61.4–50.2)		−57.6 (13.1)	−9.0	−57.1 (−65.4–47.8)		0.959	
s Y amplitude	23.4 (9.9)	1.5	21 (17–27.2)	42.3	24.3 (8.6)	3.8	22.8 (17.9–29)	35.4	0.141	

CV: coefficient of variation, s: stabilometry, SD: standard deviation, SEM: standard error of measurement, Wilcoxon test, * Not calculated if mean is negative. Bold: statistically significant.

**Table 4 ijerph-18-04940-t004:** Bodily pain and pain interfere work of SF-36 at 3-week follow-up.

**SF-36**	Physical Activity			No Physical Activity		
	Pre (n = 41)	Post (n = 41)	*p*-Value	Pre (n = 8)	Post (n = 8)	*p*-Value
**Bodily Pain**			**<0.001**			0.261
None	6 (14.6%)	19 (46.3%)		1 (12.5%)	4 (50%)	
Moderate	16 (39%)	15 (36.6%)		4 (50%)	1 (12.5%)	
Severe	19 (46.3%)	7 (17.1%)		3 (37.5%)	3 (37.5%)	
**Pain Interfere work**			**0.002**			0.9691
Not at all	20 (48.8%)	33 (80.5%)		5 (62.5%)	6 (75%)	
Moderately	12 (29.3%)	7 (17.1%)		2 (25%)	1 (12.5%)	
Extremely	9 (21.9%)	1 (2.4%)		1 (12.5%)	1 (12.5%)	

McNemar test, SF-36: 36-Item Short Form Survey. Bold: statistically significant.

**Table 5 ijerph-18-04940-t005:** Health status by EQ-5D-3L of individual at 3-week follow-up.

EQ-5D	Physical Activity			No Physical Activity		
Mean (sd)	Pre (n = 41)	Post (n = 41)	*p*-Value	Pre (n = 8)	Post (n = 8)	*p*-Value
**Mobility**			1			1
No problems	41 (100%)	40 (97.6%)		8 (100%)	8 (100%)	
Some problems	0 (0%)	1 (2.4%)		0 (0%)	0 (0%)	
I am confined in bed	0 (0%)	0 (0%)		0 (0%)	0 (0%)	
**Self-care**			1			1
No problems	41 (100%)	41 (100%)		8 (100%)	8 (100%)	
Some problems	0 (0%)	0 (0%)		0 (0%)	0 (0%)	
I am unable	0 (0%)	0 (0%)		0 (0%)	0 (0%)	
**Usual activities**			1			1
No problems	40 (97.6%)	40 (97.6%)		8 (100%)	8 (100%)	
Some problems	1 (2.4%)	1 (2.4%)		0 (0%)	0 (0%)	
I am unable				0 (0%)	0 (0%)	
**Pain/discomfort**			0.114			1
No pain	29 (70.7%)	35 (85.4%)		4 (50%)	4 (50%)	
Moderate pain	12 (29.3%)	6 (14.6%)		4 (50%)	4 (50%)	
I have extreme pain	0 (0%)	0 (0%)				
**Anxiety/depression**			**0.006**			1
Not anxious	25 (61%)	37 (90.2%)		5 (62.5%)	5 (62.5%)	
Moderately anxious	16 (39%)	4 (9.8%)		3 (37.5%)	3 (37.5%)	
Extremely anxious	0 (0%)	0 (0%)		0 (0%)	0 (0%)	
**Physical Activity**	**Pre (n = 41)**		**Post (n = 41)**			
	Mean (SD)	Median (1Q-3Q)	Mean (SD)	Median (1Q-3Q)	*p*-value	
**EQ-VAS**	79.9 (11.57)	80 (70–90)	84.5 (9.6)	85 (80–90)	**0.004**	

McNemar test; SD: standard deviation, Wilcoxon test. Bold: statistically significant.

## Data Availability

“Replication Data for The effectiveness of the Informational Manual Therapy for improving quiet standing and quality of life in Healthy Individuals”, https://doi.org/10.7910/DVN/WS6UJJ, Harvard Dataverse, V1, UNF:6:RhrJtlMrT6QM36HrfXmZLQ== [fileUNF] (accessed on 11 June 2020).

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
