# Peer review of "The Immediate Effect of Informational Manual Therapy for Improving Quiet Standing and Bodily Pain in University Population"

_ijerph, 2021, doi:10.3390/ijerph18094940_

Round 1

Reviewer 1 Report

Review of ijerph-1170220

See attached pdf file

The Immediate Effect of Informational Manual Therapy for Improving Quiet Standing and Bodily Pain in Healthy Individuals

Recommendation: This is a very interesting study utilizing a pre-post test design. Recommend some reformatting, rewording and a little more clarity for the intervention possibly utilizing pictures. Then recommend publication.

Abstract:

Lines 7-8: “A within-subjects experiment in which each 7 participant is tested first under the control condition and then under the treatment condition. “  This is actually a within subjects experiment utilizing a comparative paired analysis both before and after the intervention. Suggest re-stating this.

Line 9: “Primary outcome 9 was quiet standing” Reword: “The primary outcome…”  

Lines 12-15: Please reword: “Results: The individuals were divided into 3 age groups, 18-35 (52.6%), 35-50 (29.8%) and 51-65 (17.6%). Statistically significant differences were immediately observed after the session ended when comparing the pre-post quiet stance scores in a number of length parameters: L, Lx, Ly and stabilometry amplitude on Y-axis with eyes open and closed Significant differences were also found when testing bodily pain (SF-36) and anxiety (5Q-5D-3L).”

Lines 16-17: “Conclusion: One session of IMT 16 produces positive effects on quiet standing with eyes open and in closed conditions, and reduces 17 bodily pain and anxiety in healthy individuals.” Would reword: “One session of IMT produced positive effects when testing quiet standing with eyes open and eyes closed, as well as a significant reduction in pain and anxiety for those tested. Further research is suggested.

Introduction:

Line 24: “is a non-structural manual technique whole body approach” Would reword: “is a nonstructural manual technique using a whole-body approach with light tough.”

Lines 30-31: “Since the brain and spinal cord are contained within the cranial-pelvic system, it can have a powerful influence over a wide variety of bodily functions”

Lines 32 through 38: There are no references at all in this paragraph. This process (to some) might seem a bit too vague. Other health care professionals like myself might question the science. Your justification in this research is to show that it does scientifically work…or not. So report what you have read and state, things like, “the literature states that IMT uses cranial somatopias… etc.” And add the references. This will make someone like myself want to read it, and be curious instead of critical. J

Lines 39-45: This paragraph is much better, and there are references. Still write as though you learned this from the literature, not as though you already know it as a fact, otherwise, why do the research? Suggest rewording: “According to the literature, IMT focuses on optimizing…”

Lines 51-52: “It may be reasonably assumed that the SRM felt by palpation of the clinician derives from 51 the movement of the brain and the spinal cord;…” Again to someone like myself, this is not “reasonable” You’ve done some cool research, but writing in this definitive manner before describing your research makes the reader wonder, “why do it?” The introduction is supposed to define your research question. What is it you want to learn from this study? Instead this reads like you already know the answer. This is an easy fix. Instead write, “It has been assumed by some that the SRM felt…” and follow it with a reference.

Lines 56 to 58: This cries for a reference.

Lines 68 to 70: “Normal standing is a complex activity both mechanically and neurologically in that the apparently simple act of standing upright involves complex, dynamically regulated sensorimotor integration mechanisms that exist in the nervous system [19].” Sensorimotor integration does not exist in the nervous system alone. It’s an integration between the nervous system and the musculoskeletal system. Would reword this as “… involves complex, dynamically regulated sensorimotor integration mechanisms.”

Lines 71 to 73: “The goal of sensorimotor integration for postural control is to 71 ensure that an adequate amount of corrective torque is generated to resist the destabiliz-72 ing influence of gravity and other external perturbations” This is not the only goal of sensorimotor integration for postural control. Would change this to, “One of the goals of sensorimotor integration…”

Lines 76 to 77: “…different sensory receptors (muscle stretch, joint, Golgi tendon, foot pressure, and the tactile receptors of skin)” These are not named correctly. They should read as follows: ““…different sensory receptors (muscle stretch receptors, joint receptors such as golgitenton organs, Ruffini terminals, Pacinian corpuscles, and free nerve endings, , , and the tactile mechanoreceptors of skin)”

Lines 83 to 86: “The hypothesis of this study is that IMT has an influence on postural control, bodily pain and health status. Following along these lines, the primary focus of this study is to analyze the effectiveness of IMT as an intervention for the lessening of pain, and improvement of quiet standing balance and health status in healthy adults with somatic dysfunctions.”

Line 89: :“The study utilizes a one-group pretest-posttest design”

Lines 99-103: ‘The inclusion criteria were as follows: individuals ranging in age from 18 to 65 years old, devoid of a positive diagnosis for any form of disease which influences balance, who were not participating in any other trial, and who were free of injury in the 3 months prior to the study, with no fractures in the previous 6 months, and not having suffered any falls in the preceding month. Exclusion criteria were any individual not meeting all of the above mentioned inclusion criteria. No subject had been under any pharmacological treatment during the previous 4 weeks. 103 Every individual provided informed consent before enrolment.

Line 112: “The first level of IMT was utilized in this study.”

Line 119, “The diagnosis is performed by cranial somotopias (reflex points on the skull and face).” Tell us how many people in your study had this diagnosis. Also describe how the intervention looks to the reader. Include a picture of one of the researchers treating another researcher (don’t use the participants in your picture without IRB approval and their consent).

Data Analysis:

Nice job on the data analysis section!

Results:

Lines 169 to 177: This would look best in a table format.

Nice Flow chart!

The graphs are excellent but the “eyes open” line is difficult to see.

Suggest making this darker

Lines 218-219: “If IMT improves the precision of the fine postural system, it may prove to be an 218 important aspect in individuals with postural impairments. Further research is suggested to support these findings”

Lines 232 to 235: “The participants at the end of the IMT 232 treatment were more relaxed and their neuromuscular activity had decreased with a de-233 cline in sympathetic activity. This contributed to their improved quiet stance. In a study 234 of Tai Chi training [28], the individuals presented much better postural stability compared 235 to the general populations.” The best research articles comes full circle by referencing the articles previously discussed in the introduction, again in the discussion at the end. Either cite some of the earlier articles used again here, or take some of these new references you are using here, and also include them in your paper earlier in the introduction. The discussion isn’t the place to bring in new references. The references in the discussion should mostly be reused references from earlier in the paper. You can have one or two new ones, but the reset should be familiar to the reader.

Lines 243 through 281 Most of this should be in the intro and not in the discussion, but can then be mentioned briefly again in the discussion.

Lines 282 to 294 is good.

Lines 294 to 317: This is again information for the introduction

Lines 319-322: There is no need for a control group with a pre-post test, as each person acts as their own control, so this is not really a limitation. The major limitation is the convenience sample.

Conclusion: Excellent!

Author Response

Thank you for your suggestions and comments.

Abstract: line 7-8 I rewritten “within subjects experiment utilizing a comparative paired analysis both before and after the intervention”.

Line 9: “The primary outcome”

Lines 12-15 “The individuals were divided into 3 age groups, 18-35 (52.6%), 35-50 (29.8%) and 51-65 (17.6%). Statistically significant differences were immediately observed after the session ended when comparing the pre-post quiet stance scores in a number of length parameters: L, Lx, Ly and stabilometry amplitude on Y-axis with eyes open and closed Significant differences were also found when testing bodily pain (SF-36) and anxiety (5Q-5D-3L)”.

Lines 16-17: “Conclusion: One session of IMT 16 produces positive effects on quiet standing with eyes open and in closed conditions, and reduces 17 bodily pain and anxiety in healthy individuals.” Would reword: “One session of IMT produced positive effects when testing quiet standing with eyes open and eyes closed, as well as a significant reduction in pain and anxiety for those tested. Further research is suggested.

Introduction:

Line 24: “is a non-structural manual technique whole body approach” Would reword: “is a nonstructural manual technique using a whole-body approach with light tough.” This is done

Lines 30-31: “Since the brain and spinal cord are contained within the cranial-pelvic system, it can have a powerful influence over a wide variety of bodily functions” This is done

Lines 39-45 “Acc ording to the literature, IMT focuses on optimizing” This is done

Lines 68 to 70 “Normal standing is a complex activity both mechanically and neurologically in that the apparently simple act of standing upright involves complex, dynamically regulat-ed sensorimotor integration mechanisms” Tis is done

Lines 71 to 73 “One of the goals of sensorimotor integration…” This is done

Lines 76 to 77 “different sensory receptors (muscle stretch receptors, joint receptors such as golgitenton organs, Ruffini terminals, Pacinian corpuscles, and free nerve endings, , , and the tactile mechanoreceptors of skin)” this is done.

Lines 83 to 86 “is to analyze the effectiveness of IMT as an intervention for the lessening of pain, and improvement of quiet standing balance and health status in healthy adults with so-matic dysfunctions” This is done

Line 89 “The study utilizes a one-group pretest-posttest design” This is done

Lines 99-103: ‘The inclusion criteria were as follows: individuals ranging in age from 18 to 65 years old, devoid of a positive diagnosis for any form of disease which influences balance, who were not participating in any other trial, and who were free of injury in the 3 months prior to the study, with no fractures in the previous 6 months, and not having suffered any falls in the preceding month. Exclusion criteria were any individual not meeting all of the above mentioned inclusion criteria. No subject had been under any pharmacological treatment during the previous 4 weeks. 103 Every individual provided informed consent before enrolment” this was done

Line 112: “The first level of IMT was utilized in this study.” this was done

We have done three pictures.

Results:

Lines 169 to 177: This would look best in a table format. This was done

Lines 218-219: “If IMT improves the precision of the fine postural system, it may prove to be an important aspect in individuals with postural impairments. Further research is suggested to support these findings” this was done

Lines 319-322: The major limitation is the convenience sample. This was done.

The graphics are darker.

We have cited some articles used in the introduction in the discussion.

Reviewer 2 Report

This manuscript analysed the Immediate Effect of Informational Manual Therapy for Improving Quiet Standing and Bodily Pain in Healthy Individuals, which is meaningful data. I think it would be a more clear study if the following parts were revised and supplemented.

Page 1 lines 32-38 to need include references related with this declaration
Add  a figure according of this Intervention for clarify the readers this method.
Discussion: There is a lack of quantity and quality of discussion to understand the results of this study.  It is recommended that the author’s view and analysis (including mechanism analysis) of the results be added rather than a simple comparison to previous studies.

Author Response

Thank you for your comments and suggestions.

Page 1 lines 32-38 to need include references related with this declaration. We have included more references.

Add a figure according of this Intervention for clarify the readers this method. We have made three pictures.

Discussion: There is a lack of quantity and quality of discussion to understand the results of this study.  It is recommended that the author’s view and analysis (including mechanism analysis) of the results be added rather than a simple comparison to previous studies.

We have included an analysis of the results in the discussion. 

Reviewer 3 Report

The ICC should be indicated and SEM and CV should be provided for the reliability assessments in the primary and secondary outcomes . (e.g., Hopkins, Sports Med, 2000).

Author Response

Thank you for your comments and suggestions.

The ICC, SEM and CV were done.

Reviewer 4 Report

Dear Authors 

Thank you for the submission of the manuscript titled “The Immediate Effect of Informational Manual Therapy for Improving Quiet Standing and Bodily Pain in Healthy Individuals", to my review.

Dear Authors
I realize that authors have many journals to consider when they want to publish their work, so I appreciate your interest in "Int. J. Environ. Res. Public Health"; I am very sorry not to be able to write in a more positive way. It is evident that you have put a great deal of effort into this project and I want to praise your efforts, but unfortunately, the actual contribution from your paper to scientific literature is not clear or strong. The manuscript as currently written not suggests that it might be suitable for sharing information about this "manual therapy method", the  study you reported is not representative to state with certainty your conclusions. Missing more strong evidence in your study.
I am very sorry not to be able to write in a more positive way but I should like to thank you for give me an opportunity to consider this work for publication.
Best Regards

Author Response

Thank you for your comments.

The manuscript was rewritten.

Reviewer 5 Report

Thank you for letting me review the manuscript entitled "The immediate effect of informational manual therapy for improving quit standing and bodily pain in healthy individuals"

The manuscript is interesiting and covers and specific topic. However I have few suggestions for the authors: 

Methods: 

  • The explanation of the intervention is not clear. This subheading needs further explanation.
  • Also, the patients included in the study were healthy individuals with no symptoms, so how did you detect the dysfunctions? How did you know that the patient need to harmonize the cranial-pelvic systems?
  • Concerning the outcome measures, the included patients were healthy individuals. Why did you decide to measure bodily pain and quality of life? I supossed the patients were already pain-free.

Discussion and conclusion. 

- The manuscript did not incuded a control group or a sham group, moreover the patients were their own controls. According to these findings, the changes cannot be directly linked to the therapy. The results achived could be due to the placebo effect. Please reflect this in the discussion and conclusion part. 

Author Response

Thanks you for your comments and suggestions.

Methods:

The explanation of the intervention is not clear. This subheading needs further explanation.

The intervention was rewritten.

Also, the patients included in the study were healthy individuals with no symptoms, so how did you detect the dysfunctions? Because when the therapist checks the “fuses” is stopped, there is not the normal movement. How did you know that the patient need to harmonize the cranial-pelvic systems? The hand therapist is trained to feel the subtle movement, if any fuse is wrong the therapist feels it, the secondary respiratory movement is not harmonic.

Concerning the outcome measures, the included patients were healthy individuals. Why did you decide to measure bodily pain and quality of life? I supposed the patients were already pain-free. We supposed that the participants were university population, students and professors and there were very stressed and it causes alterations in the body. It is explained in the manuscript “Our participants had stated positive effects on bodily pain and anxiety. They were healthy adults and were not diagnosed with any disease. The majority were university students and professors. However, about 85% of them presented bodily pain and 38% anxiety. This was in line with the results of a meta-analysis of Quek et al. They showed that a 33.8% of medical students suffered anxiety. Backåberg et al. revealed that university education has an impact in musculoskeletal symptoms. Our results suggested that this could be an important factor in relation to patients with various diseases and individuals suffering from stress. With regards to stress, the cerebral cortex sends signals to the autonomous nervous system, turning the sympathetic to becoming active. This results in the increase of cardiac frequency and changes in the variability of the frequency, reducing the flow of peripheral blood and the flow of renal blood. This in-creases the blood pressure and vascular resistance”.

Discussion and conclusion.

- The manuscript did not included a control group or a sham group, moreover the patients were their own controls. According to these findings, the changes cannot be directly linked to the therapy. The results achieved could be due to the placebo effect. Please reflect this in the discussion and conclusion part.

Yes, we are agreeing with you but for this reason we choose the stabilometry by a force platform to measure quiet standing.  This device is an objective measure and in our opinion the possible placebo effect is not recorded.

Round 2

Reviewer 4 Report

Dear Authors
thank you for rewriting the text and for trying to respond to all comments, but although the study appears well written and conducted, it is supported by unscientific foundations or supported by strong scientific evidence. The scientific perplexities remain too high. Moreover, reference number 2 is not a bibliographic reference, 8 references are by Bordoni et al., that confirmed that the study not supported of scientific literature in different fields. Changing the title from healthy subjects to university subjects does not improve the conceptual bias that studies pain in healthy subjects.
I am very sorry not to be able to write in a more positive way but I should like to thank you for give me an opportunity to consider this work for publication. Best Regards.

Author Response

Respons 2.

Thank you for your suggestions and comments.

We have removed Bordoni et al. CUREUS journal references and they have been changed by others.

The reference 2 has also been changed. 

Best Regards,